# Percolation on feature-enriched interconnected systems

Oriol Artime [1✉] & Manlio De Domenico [1✉]

Percolation is an emblematic model to assess the robustness of interconnected systems when some of their components are corrupted. It is usually investigated in simple scenarios, such as the removal of the system's units in random order, or sequentially ordered by specific topological descriptors. However, in the vast majority of empirical applications, it is required to dismantle the network following more sophisticated protocols, for instance, by combining topological properties and non-topological node metadata. We propose a novel mathematical framework to fill this gap: networks are enriched with features and their nodes are removed according to the importance in the feature space. We consider features of different nature, from ones related to the network construction to ones related to dynamical processes such as epidemic spreading. Our framework not only provides a natural generalization of percolation but, more importantly, offers an accurate way to test the robustness of networks in realistic scenarios.

[1] Center for Information and Communication Technology, Fondazione Bruno Kessler, Povo, TN, Italy. ✉email: oartime@fbk.eu; mdedomenico@fbk.eu

Classical percolation is a toy model in which one deletes nodes from a low-dimensional regular lattice and computes different properties, statistical and geometrical, of the remaining isolated clusters. Although firstly considered as a gelation problem in the context of macromolecules[1], it is not until the development of the theory of critical phenomena[2] that percolation drew a great deal of attention to the physics community. The reasons were, at least, twofold. On one hand, percolation provided stimulating theoretical challenges, considered one of the simplest models displaying a phase transition, without any need of introducing dynamics or thermodynamics quantities, as it occurs in the Ising model[3], for instance. On the other hand, percolation was flexible enough in its definition to be mapped to many diverse problems, such as the dielectric response of inhomogeneous materials[4], epidemiology[5] or flows in porous media[6], among many other[7,8].

The interest in percolation is renewed with the advent of modern network theory[9]. A network, broadly construed, is a set of nodes with arbitrary connections among them, contrary to lattices, that are regular structures embedded in spaces of finite dimension, with all of their nodes having the same number of edges, i.e., same *degree*. In this context, the fraction of removed nodes are usually thought of as failures or attacks, and the largest connected component after the perturbation has a functional interpretation assumed to be the part of the network that is still operative. Therefore, percolation in this type of topologies has brought a deeper understanding of the robustness and resilience of real-world networked systems[10–13], as well as, from a fundamental perspective, it has provided new analytical techniques[14,15] and interesting phenomenology from the standpoint of statistical physics[16–19].

Since in networks the degree of nodes is distributed heterogeneously, one can exploit this fact to devise new physically meaningful removal strategies, such as targeting nodes from higher to lower degree[14,20]. These interventions on the networks are called *attacks* since they are intentionally performed using some a priori information. Studying percolation based on degree attacks elucidates the role played by large degree nodes, the hubs, on the network robustness. For instance, graphs with long-tailed degree distributions are very weak to hub removal, that is, by removing a very small number of hubs the network is broken into many small components. This has catastrophic consequences for the security of real-world networks since many of them display such degree distributions[21].

The degree is the most basic centrality measure in complex networks. However, there exist a plethora of alternatives to assess the importance of a node within a graph[22–24], and accordingly, we can test the importance of these variables on the network robustness by performing attacks based on them[25–27] and evaluate how far is each attack strategy from being optimal[28]. Moreover, nodes could be characterized by non-topological properties as well, such as age[29], biomass[30], or bank credibility[31]. Hence, similar network attacks can be implemented to test percolation properties of the system when a group of nodes with certain characteristics is removed, for instance, those users of online social platforms generating or spreading hateful content[10].

Taking into account that in many relevant situations singular information is accessible at a node level, it is desirable to have a method to quantify the impact of intervening in the network following alternative protocols based on these non-topological features. We tackle therefore the challenge of developing a percolation framework that accounts for both topological and non-topological information simultaneously. The latter element will be considered as node metadata, what we call the *features*. We generalize standard message-passing methods by introducing a joint degree-feature probability density function on the network. Several percolation quantities, such as the critical point or the size of the giant component are computed. We check the validity of our theory by confronting the analytical estimates with synthetic and real-world networks, finding an excellent agreement.

The rest of the paper is organized as follows. We first motivate the usefulness of feature-enriched percolation by presenting some examples of real-world networks with different degree-feature correlations and proposing virtual dismantling experiments on them. Next, the model is presented and we work out the analytical expression for the size of the giant component in terms of a generic degree-feature joint probability function. We then confirm the analytical predictions in synthetic networks, discussing separately the cases of uncorrelated and correlated degree-feature distributions. We also test the validity of our theory in random geometric graphs (RGGs), which are known to be highly spatially correlated, and therefore, message passing methods may fail to accurately predict the percolation point. A final section is devoted to the very interesting case in which the features are related to variables coming from dynamic processes running on top of networks. We investigate these latter cases in both synthetic and real-world networks. We close the article by drawing conclusions.

## Results

**Empirical evidence of non-trivial feature distributions**. In this section, we report different patterns of feature distributions in real-world complex networks. The chosen examples are arbitrarily selected based on our biases, but they are not, by no means, an exception. Indeed, when collecting real data to construct network structures, most of the times the nodes have some associated properties, or *metadata*, that individually characterize them beyond their degree.

The first example corresponds to a board interlock network, i.e., a bipartite system of directors and companies. Links between them exist whenever a director holds a seat on the corporate board of a company, and nothing prevents a director to sit on more than one boardroom. We regard the feature as the age of the members of the board. We see that, in this case, the average age of the directors is uncorrelated with respect to their degree, see Fig. 1a. Nevertheless, the spread of the distribution as a function of the degree is not constant. Hence, feature-enriched percolation in this example can help reveal the role played by directors of a certain age on the global connectivity structure of the system.

The second example belongs to the context of crowdsourced creation of cultural content. We take the snapshot of the current version of Wikipedia, in which nodes are articles and edges are the hyperlinks among them. Moreover, for each node we keep track of the number of revisions that it has suffered since its creation, and how many unique users have participated in these edits. We see that the correlation between the degree of a Wikipedia page and its number of unique editors is positive and heterogeneously distributed, see Fig. 1b. With the framework of feature-enriched percolation, we are able to remove nodes with a certain degree-feature pattern, thus, for example, one can be interested to assess how the navigability across this knowledge corpus is modified under the removal of well-connected articles that do not show enough levels of collaborative edition, i.e., that have been written by too few users.

As a final example, we present a system that displays a negative correlation among degree and feature, see Fig. 1c. It corresponds to misinformation propagation in the online social platform Twitter. We take a sample of messages within a two-week time window during the COVID-19 pandemic and consider only messages that share an url in the text. The nodes are users, where the degree is their number of followers, and the feature is the number of fake news that they have posted. We see that the average number of fake news tends to decrease as the visibility of Twitter accounts is higher, varying broadly in this case as well. Feature-enriched percolation can be helpful, for example, to shed light on the problem of how to

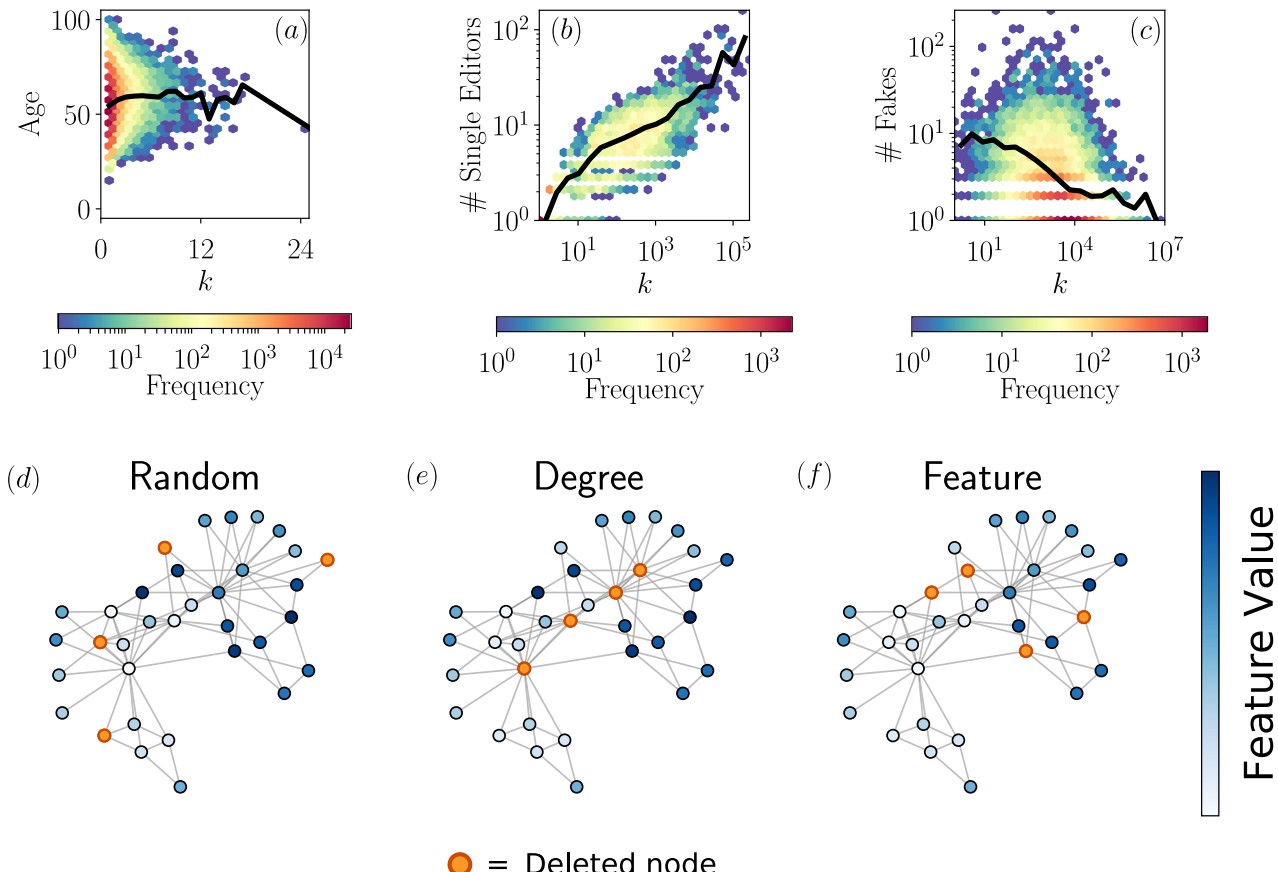

**Fig. 1 Empirical evidence of different feature-degree patterns and sketch of different types of removal protocols in the percolation process.**
**a** Correlation between the degree of a director of a company (number of companies in which she participates) and her age. Data from[67]. **b** Wikipedia pages show a positive correlation between the hyperlinks present in an article (out-degree) and the number of unique editors that have modified such article. Data from an updated version of the ref. [68]. **c** Negative correlation between the number of followers and the number of fake news posted by a user. The subset of data taken from[69], comprising worldwide users during the first two weeks of April 2020. The black solid lines correspond to the mean value of the data. In the bottom row, we show how different interventions can affect the network. In the random case (**d**) nodes are chosen equiprobably to be deleted. In the degree case (**e**), the removed nodes are those with the largest number of connections. In the feature case (**f**), the selected elements are those with the largest value of the feature (the darkest blue). In this example, the values of the feature have been assigned manually for convenience and the feature vector **F** has been considered unidimensional.

guarantee the information spreading while cutting off users that systematically share dysfunctional and harmful content.

**The model**. Let us take the ensemble of networks generated via the configurational model[9], with a degree sequence obtained from a degree distribution $p_k$. We assume that each node is characterized by its degree $k$ and by a set of features $\mathbf{F} = (F_1, F_2, ..., F_M)$, with $M \in \mathbb{N}$. The degree distribution is considered discrete and the feature distribution can be considered either discrete $p_{\mathbf{F}}$ or continuous $p(\mathbf{F})$. Throughout the article, we shall assume that features are continuous, but reproducing the results for a discrete domain is straightforward. The joint probability is indicated by $P(k, \mathbf{F})$. We define the occupation probability $\phi_{k,\mathbf{F}}$ as the probability that a node with degree $k$ and feature values $\mathbf{F}$ within the interval $[\mathbf{F}, \mathbf{F} + d\mathbf{F}]$ have not been removed from the original network. Our goal is to find the position of the critical point and the size of the giant connected component $S$ as a function of the parameters of $P(k, \mathbf{F})$ and $\phi_{k,\mathbf{F}}$[32].

Let $u$ be the average probability that a node with $k$ connections and feature values $\mathbf{F}$ does not belong to the giant component via one of its neighbors. The probability that it does not belong to the giant component is $u^k$. Thus, the probability that the node belongs to the giant component due to a neighbor state is $1 - u^k$ and has to be multiplied by $\phi_{k,\mathbf{F}}$, which determines whether or not

the node itself is present in the network. Averaging this quantity over degrees and features we obtain $\int d\mathbf{F} \sum_k \phi_{k,\mathbf{F}} P(k, \mathbf{F})(1 - u^k)$, where $\int d\mathbf{F}$ is an $M$-dimensional definite integration over the elements of the feature vector. This is identified as the average probability of finding a node in the giant cluster, or equivalently, the fraction of nodes in the giant component. Therefore

$$S = g_0(1) - g_0(u), \qquad (1)$$

where the generating function is

$$g_0(z) \equiv \int d\mathbf{F} \sum_k \phi_{k,\mathbf{F}} P(k, \mathbf{F}) z^k. \qquad (2)$$

To solve Eq. (1) we need to obtain an expression for $u$. This can be achieved by writing a self-consistent equation that has two contributions. On the one hand, a neighbor might not be in the giant component because it has been deleted, which occurs with probability $1 - \phi_{k,\mathbf{F}}$. On the other hand, if the neighbor has not been removed, it should not belong to the giant component via any of its other $k_e$ neighbors. $k_e$ is called the excess degree distribution, and it is equal to $k_e = k - 1$. This happens with

probability $\phi_{k,\mathbf{F}}u^{k_e}$. Averaging over the distributions, we get

$$u = \int d\mathbf{F} \sum_{k_e=0}^{\infty} Q(k_e, \mathbf{F})\left[(1 - \phi_{k_e+1,\mathbf{F}}) + \phi_{k_e+1,\mathbf{F}}u^{k_e}\right]$$
$$= 1 - \frac{1}{\langle k \rangle}\int d\mathbf{F}\sum_{k=1}^{\infty}kP(k,\mathbf{F})\phi_{k,\mathbf{F}}(1 - u^{k-1}). \quad (3)$$

Here $Q(k_e, \mathbf{F})$ is the excess degree-feature distribution, which is normalized and verifies $Q(k_e, \mathbf{F}) = (k_e + 1)P(k_e + 1, \mathbf{F})/\langle k \rangle$, with the mean degree computed as $\langle k \rangle = \int d\mathbf{F}\sum_k kP(k, \mathbf{F})$. Introducing the generating function as

$$g_1(z) \equiv \frac{1}{\langle k \rangle}\int d\mathbf{F}\sum_{k=1}^{\infty}kP(k,\mathbf{F})\phi_{k,\mathbf{F}}z^{k-1}, \quad (4)$$

Eq. (3) simplifies to

$$u = 1 - g_1(1) + g_1(u). \quad (5)$$

We readily obtain the size of the giant component by plugging the solutions of Eq. (5) into Eq. (1). Notice that $u = 1$ is always a solution for Eq. (1), corresponding to $S = 0$. To observe the percolating structure we need to use the other solution, which appears when the condition $1 = g'(u)|_{u=1}$ is met. The existence of an analytical expression will depend on $P(k, F)$ and $\phi_{k,F}$, otherwise, it can always be solved numerically.

Notice that the relation between generating functions held in ordinary percolation is valid here as well, i.e., $g_1(z) = \partial_z g_0(z)/\langle k \rangle$. It is important to also note that our generating functions, although including the $M$-dimensional integral in the definition, depend only on one variable since the feature enrichment does not add any new information in terms of connectivity. Therefore, our framework should not be taken equivalent, for instance, to the study of percolation in graphs with colored edges[33], in general multilayers[34], in networks with multi-type nodes[35] or in interdependent systems[36], where multivariable generating functions are common.

The occupation probability $\phi_{k,\mathbf{F}}$ allows us to understand the role played by certain values of degree and/or features in the connectivity of the network. The classical percolation, where nodes are removed in a uniformly random fashion, is recovered by selecting a constant function $\phi_{k,\mathbf{F}} = \phi \in [0, 1]$. The case of removing the most connected nodes is recovered by setting $\phi_{k,\mathbf{F}} = \theta(-(k - k_0))$, being $\theta(\cdot)$ the Heaviside step function and $k_0$ a threshold such that all nodes with degree larger than it are removed. Similarly, one can apply the same arguments in the feature space, and study the case in which all nodes with a feature larger than a threshold are deleted $\phi_{k,\mathbf{F}} = \theta(-(\mathbf{F} - \mathbf{F_0}))$. These three examples are sketched in Fig. 1.

**Applications.** To illustrate and check the validity of the theory, we investigate several examples. For the sake of simplicity we focus on unidimensional feature vectors, i.e., $\mathbf{F} = F$. First we address the case of independent degree and feature, i.e., $P(k, F) = p_k P(F)$. We then move to consider joint distributions which are positively and negatively correlated. These latter cases leave the nature of the feature undetermined. In this section, though, we also address problems in which the features are related to the distance in a geometrical space and to dynamical processes evolving on top of the network.

**Independent case.** Let us consider a network with degree distribution and feature distribution

$$p_k = (1 - a)a^k,$$
$$p(F) = (\alpha - 1)F^{-\alpha}, \quad (6)$$

where $k = 0, 1, 2,...$, and $F \in [1, \infty)$ and $\alpha > 1$. We take as occupation probability $\phi_F = \theta(-(F - F_0))$, that is, all nodes with feature $F > F_0$ are removed. In this case, the generating functions are

$$g_0(u) = (1 - F_0^{1-\alpha})\frac{1 - a}{1 - au},$$
$$g_1(u) = (1 - F_0^{1-\alpha})\left(\frac{1 - a}{1 - au}\right)^2. \quad (7)$$

It does not exist a closed expression for the size of the giant component, but it is straightforward to obtain a numerical solution. In Fig. 2 we compare the numerical solutions with the actual process of percolation and we see that the agreement is excellent. Figure 2a displays $S$ against the parameter characterizing the topology and we observe, as one could expect, that the network needs to be dense enough to observe the emergence of the giant component. In Fig. 2b we plot the dependence of $S$ on the parameter characterizing the feature distribution. In this case, we see that if the feature distribution does not decay fast enough, no giant component is possible. Notice that in Fig. 2b the critical point is <2, i.e., the feature distribution has a diverging mean value. This case might seem extreme or unrealistic, but the critical point $\alpha_c$ can be located within the interval $[2, 3]$—a much more common case—if $a$ is small enough. This evinces the important role that the feature distribution may play for the robustness of a network to, for instance, attacks that are feature-based. For completeness, we give the values of the critical points, that in this case have a closed expression, namely

$$a_c = \frac{1}{3 - 2F_0^{1-\alpha}},$$
$$\alpha_c = 1 - \frac{\log\left(\frac{3a-1}{2a}\right)}{\log(F_0)}. \quad (8)$$

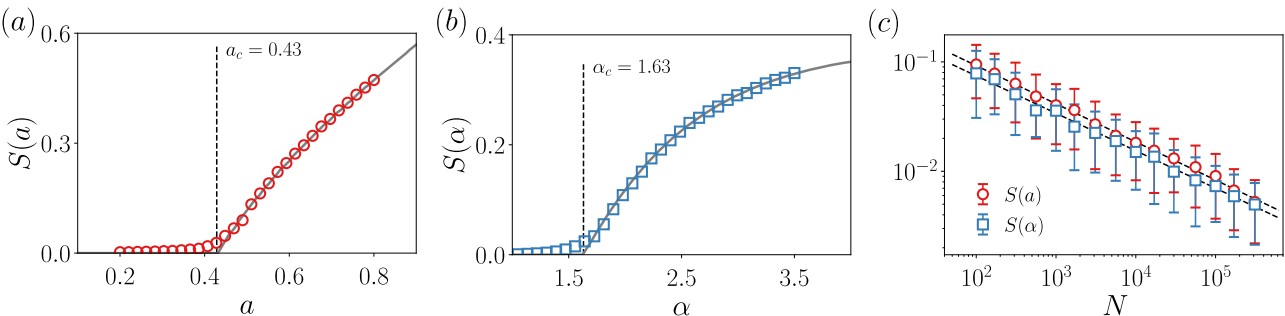

**Fig. 2 Size of the giant component for the independent degree-feature case.** Solid lines are computed from the theory, the markers come from simulations. In (**a**), the control parameter $a$ is related to the topology, while in (**b**) the control parameter $\alpha$ is related to the feature distribution, see Eq. (6). Vertical lines indicate the value of the critical point computed from the theory, see Eq. (8). System size is $N = 3000$ and each point is computed by averaging over 100 independent realizations. In (**c**), the size of the giant component with error bars as a function of the system size, together with the fits to the data.

To show that the good agreement between theory and simulations extends to other topologies, the same analyses conducted above are presented in Supplementary Notes 1 and 2 for the Erdős–Rényi and scale-free network models.

We investigate the universality class of the percolation process for independent feature-degree distributions, in order to figure out whether the introduction of features modifies the critical properties of mean-field percolation. Here we understand mean-field as the behavior of classical percolation in regular lattices of dimension $d \geq d_c = 6$. The critical properties of complex networks, which are infinite-dimensional entities, might not be the same as mean-field, if, for instance, the underlying degree distribution is power-law[16] or the removal process is modified[37]. Note that in feature-enriched percolation we will necessarily have two parameters, one controlling the topology and another the features. Hence, for the size of the giant component, we write $S(a) \sim (a - a_c)^{\beta_a}$ and $S(\alpha) \sim (\alpha - \alpha_c)^{\beta_\alpha}$, being $\beta_a$ and $\beta_\alpha$ the corresponding critical exponents. (To be accurate, $S(a)$ also depends on $\alpha$ and $S(\alpha)$ depends on $a$ too, although when studying the critical behavior, they are taken as constants, hence we do not write them for the sake of clearness.) In Supplementary Note 3 we analytically show that $\beta_a = \beta_\alpha = 1$, i.e., it takes the mean-field value. To find other critical exponents, we can employ the finite-size scaling hypothesis

$$\begin{aligned} S(a, N) &= N^{-\beta_a/\bar{\nu}_a} \mathcal{F}\left(|a - a_c| N^{1/\bar{\nu}_a}\right), \\ S(\alpha, N) &= N^{-\beta_\alpha/\bar{\nu}_\alpha} \mathcal{G}\left(|\alpha - \alpha_c| N^{1/\bar{\nu}_\alpha}\right). \end{aligned} \quad (9)$$

Thus, the critical exponent $\bar{\nu}$ can be immediately found by fitting the resulting power law of the size of the largest connected component against the system size, at the critical point. Note that since networks are infinite-dimensional, $\bar{\nu} = d_c \nu$[29], where $d_c = 6$ is the percolation upper critical dimension and $\nu$ the typical critical exponent associated with the correlation length. The results are shown in Fig. 2c, finding that $\beta_a/\bar{\nu}_a = 0.347 \pm 0.015$ and $\beta_\alpha/\bar{\nu}_\alpha = 0.342 \pm 0.021$. The values of $\bar{\nu}_a$ and $\bar{\nu}_\alpha$ agree well with the mean-field percolation exponent $\bar{\nu} = 3$. We arrive at the same conclusion by employing data collapse based on the finite-size scaling relations, see Supplementary Note 4. Therefore, we conclude that the critical properties are the same as the mean-field percolation process, even though if the feature distribution is scale-free[16].

**Positively correlated case**. Let us consider now the more interesting and realistic case of a joint distribution in which feature and degree are not separable. We take one of the simplest scale-free distributions that are positively correlated,

$$P(k, F) = \frac{\mathcal{Z}}{(k + F)^{2+\alpha}}, \quad (10)$$

with $k \in \mathbb{N}$, $F \in [1, \infty)$, $\alpha > 1$, and normalization constant

$$\mathcal{Z} = \frac{1 + \alpha}{\zeta(1 + \alpha) - 1}. \quad (11)$$

$\zeta(\cdot)$ is the Riemann zeta function[38]. The correlation is positive because the nodes with a high degree tend to have larger values of the feature, a property that can be easily checked by computing the conditional average degree $\langle k(F) \rangle$. The distribution (10) has been considered before for instance in the context of transport properties on weighted networks[39] or in temporal correlations of dynamical processes[40].

In practical terms, to assign the values of the degree and the feature in the simulations, we first construct the network from the configurational model with a degree sequence drawn from $p_k = \int dF P(k, F)$. Then, depending on the degree of each node, we draw a random variable from the conditioned distribution $P(F|k)$. Randomized versions of the system are possible either by randomly shuffling the feature of the nodes in the correlated network or by constructing a new network from $p_k$ and assign features from $P(F)$.

Considering again the removal of all nodes with feature value above a threshold $F_0$, the generating functions are

$$\begin{aligned} g_0(u) &= \left[\frac{\Phi(u, \alpha + 1, 2) - \Phi(u, \alpha + 1, F_0 + 1)}{\zeta(\alpha + 1)}\right] u, \\ g_1(u) &= \frac{1}{\zeta(\alpha) - \zeta(\alpha + 1)} \big[\Phi(u, \alpha, 2) - \Phi(u, \alpha + 1, 2) \\ &\quad - \Phi(u, \alpha, F_0 + 1) + F_0 \Phi(u, \alpha + 1, F_0 + 1)\big], \end{aligned} \quad (12)$$

where $\Phi(\cdot, \cdot, \cdot)$ is the Lerch transcendent function[38]. A closed expression for the size of the largest connected component exists, although it is long and not very enlightening, so we do not report it here.

The theoretical predictions are compared with the simulations in Fig. 3a, finding an excellent agreement. Another curve is also displayed, corresponding to the randomized version of the correlated model Eq. (10), and that serves to single out the role of the degree-feature correlations in the behavior of the size of the giant component. It is immediate to observe that the critical point and the size of the giant component are smaller for the correlated case than for the randomized case. This holds true for all feature thresholds, being the separation among both the critical points and the $S$ more accentuated as $F_0$ decreases. The rationale behind this is the following: the marginal feature probability $P(F)$ is the same for

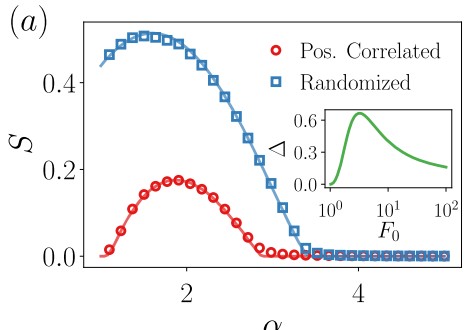
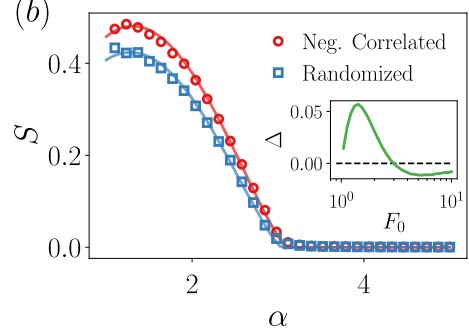

**Fig. 3 Percolation in correlated degree-feature networks.** Size of the giant connected component for (**a**) the positively correlated distribution Eq. (10) and for (**b**) the negatively correlated distribution Eq. (14), and their randomized counterparts. All nodes with feature $F \geq F_0 = 3$, are removed. Points are the results from simulations and the solid line comes from the theory. Network sizes are $N = 2^{16}$ and each point is averaged over 100 realizations. In the inset of the panels, we display $\Delta(F_0)$, corresponding to the area between the randomized and correlated curves and defined in Eq. (13), as a function of the feature threshold. The integral has been computed in the region $\alpha \in (1, 5]$.

both cases, so for a given threshold $F_0$ the same fraction of nodes is removed, although the chances of hitting a low-degree node are much larger in the randomized case than in the correlated case, due to, precisely, the degree-feature correlation. Since targeting hubs it is a very fast way of dismantling a network, it is natural to observe a smaller critical point and a smaller size of the giant component. A quantitative way to measure the deviations caused by degree-feature correlations with respect to the uncorrelated scenario is to compute the area between both curves,

$$\Delta(\mathbf{y}) = \int_{\mathbf{x}_{min}}^{\mathbf{x}_{max}} d\mathbf{x} \left[ S_{rand}(\mathbf{x}, \mathbf{y}) - S_{corr}(\mathbf{x}, \mathbf{y}) \right], \quad (13)$$

where we have split the dependency of the order parameter into two subsets of variables, for the sake of generality. The sign of $\Delta$ gives information on whether the correlations reduce the robustness (positive sign) or enhance it (negative sign). In the inset of Fig. 3a, we show $\Delta(F_0)$, indicating that the behavior depicted in the main panel, and discussed above, is hold for any value of the feature threshold $F_0$. Surprisingly, $\Delta(F_0)$ is non-monotonic and presents an optimal value of the threshold for which the correlations induce the largest robustness reduction.

Figure 3a also shows a striking peculiarity: the non-monotonicity of the giant component and its approach to the non-percolating regime $S = 0$ when $\alpha \to 1$. In Supplementary Note 5, we give an explanation for this behavior and show that there are no critical properties in the vicinity of $\alpha = 1$. This approach to the non-percolating phase is a genuine effect of the degree-feature correlation that would be missed if the correlation is disregarded. In other words, the robustness of this positively correlated system under the attack to its most prominent nodes in the feature space would be overestimated if correlations are ignored. In Supplementary Note 6 (see Supplementary Movies 1 and 2 as well) we also address the critical properties of this positively correlated network model, where we show that it shares the same critical exponents as the mean-field percolation.

**Negatively correlated case.** Contrary to the last section, there are certain situations in which peripheral, low degree nodes might be the ones carrying the largest feature values. Again, we model this scenario with an ad hoc negatively correlated joint distribution, one of the simplest displaying power-law behavior[39]:

$$P(k, F) = \frac{\mathcal{Z}}{(kF + 1)^{\alpha+1}}, \quad (14)$$

with $\alpha > 1$ and normalization constant

$$\mathcal{Z} = \alpha \left( \sum_{k=1}^{\infty} \frac{1}{k(k+1)^{\alpha}} \right)^{-1}. \quad (15)$$

When the exponent $\alpha$ is an integer, $\mathcal{Z}$ can be expressed as a combination of polygamma functions, but we were not able to find a closed expression for a non-integer exponent. Anyway, the sum is immediate to compute in any software for numerical calculations. The anticorrelation can be appreciated in the decaying dependence of the conditioned mean degree $\langle k(F) \rangle$ with the feature value.

The generating functions corresponding to Eq. (14), and for the same occupation probability as the other examples, are

$$g_0(u) = \frac{\mathcal{Z}}{\alpha} \sum_{k=1}^{\infty} \left[ \frac{1}{(1+k)^{\alpha}} - \frac{1}{(1+kF_0)^{\alpha}} \right] \frac{u^k}{k}$$

$$g_1(u) = \frac{1}{\zeta(\alpha)} \left[ \Phi(u, \alpha, 2) - F_0^{-\alpha} \Phi\left(u, \alpha, 1 + \frac{1}{F_0}\right) \right]. \quad (16)$$

The comparison between the theoretical predictions and the simulations for the negatively correlated distribution is given in Fig. 3b, both for the correlated distribution and its randomized version, finding again a very good agreement. We observe that in

this case the behavior is reversed with respect to what is found in the positively correlated case. The critical point and the size of the giant component for the anticorrelated networks are slightly higher than the one for the randomized counterparts. However, surprisingly this behavior depends on the feature threshold employed. Depending on the $F_0$, we might be in regimes in which disregarding the negative degree-feature correlations leads to an underestimation or to an overestimate of the robustness, see the inset of Fig. 3b. Similarly to the positively correlated case, $\Delta$ is non-monotonic and there is an intermediate threshold value for which the underestimation is maximum, but increasing $F_0$ even more we find the point at which the overestimation is maximum. This evinces the non-trivial phenomenology that arises even in these simple cases of ad hoc correlations.

**Features related to the network construction.** Let us now consider a case where the degree and feature distributions are not imposed exogenously but they emerge naturally from the network construction process. As an example, we consider RGGs, which are especially useful to model situations in which there is some kind of physical contact or proximity between the units of the system, and they find applications in areas as diverse as wireless sensor architectures[41], population dynamics[42], or consensus formation[43], to name but a few. Even though percolation has been widely studied in RGGs in the mathematical literature (see, e.g., Refs. [44,45]), critical points are known up to some bounds[46], and because of the strong topological correlations of RGGs, tree-like approximations, in general, are not as accurate as in the case of infinite-dimensional networks. On this basis, we will need to quantify the discrepancy between theory and simulations and how it scales with the link density.

We consider RGGs composed of $N$ nodes, placed uniformly at random on $[0,1)^2$ with periodic boundary conditions. Two nodes are connected if they are within a distance $r$. We take the feature as the distance between a node and its closest neighbor $d_{min}$, i.e., $F \equiv d_{min}$, and we delete those nodes that have the nearest neighbor at a distance smaller than a certain threshold $r_0 > 0$, i.e., $\phi_F = \theta(F - r_0)$ (see Fig. 4a). This scenario can be relevant, for instance, in spatial ecological models in which there is a competition for resources[47,48]. Note that this particular choice of the occupation probability is made for illustrative purposes, and without much effort, one could envision other relevant scenarios, for example, with $\phi_F = \theta(-(F - r_0))$ or that take the distance to the furthest neighbor $d_{max}$ as the feature. In all these mentioned cases, the generating functions can be calculated.

To compute the generating functions we first need the joint probability function. Its calculation is detailed in Supplementary Note 7. Setting a maximum bound $k_{max} = N - 1$ in the degree distributions, the generating functions are

$$g_0(u) = \left[ 1 + \pi r^2(u-1) - \pi r_0^2 u \right]^{N-1} - \left[ 1 - \pi r^2 \right]^{N-1},$$

$$g_1(u) = \frac{(r - r_0)(r + r_0)}{r^2} \left[ 1 + \pi r^2(u-1) - \pi r_0^2 u \right]^{N-2}. \quad (17)$$

The analytical calculations are compared with simulations in Fig. 4b. We keep the radius of interaction $r$ constant and discuss the results as a function of the number of nodes in the network. First of all, we see that the position of the percolation point is inversely proportional to $N$. This behavior is somehow expected: the larger the system size, the denser the network, so the probability of nodes to have neighbors at distance smaller than $r_0$ becomes high, hence the rapid network dismantling. Moreover, we see that when the system size is low, the theoretical results systematically overestimate the value of the giant component. Note that this is not a finite-size effect as it occurs in other systems displaying phase transitions, but inherent limitations of

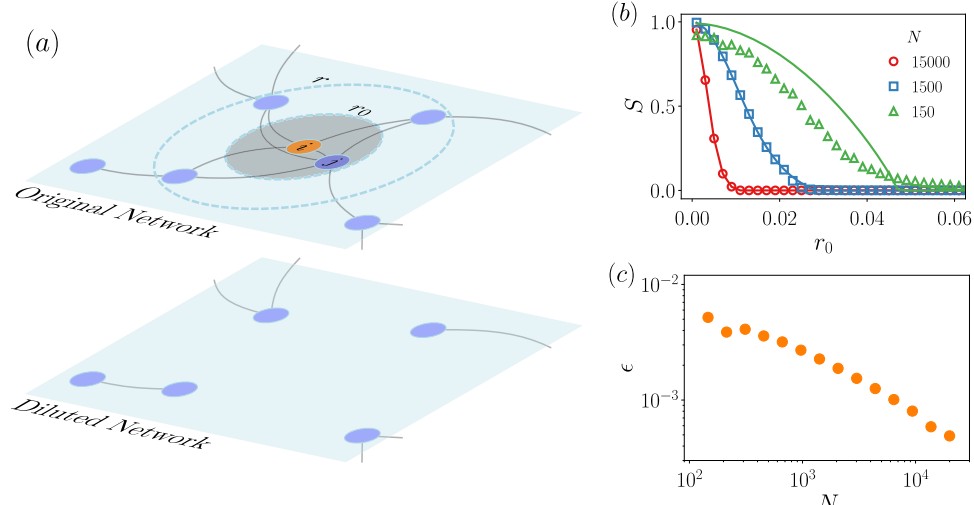

**Fig. 4 Feature-based percolation in random geometric graphs.** In (**a**), a sketch showing how the removal of nodes proposed in the main text works: if a node $i$ (central orange marker in this case) has its closest neighbor at a distance smaller than $r_0$ (shaded area), then $i$ is removed. This removal condition is checked synchronously for all nodes at once, hence node $j$ in the sketch is also removed because, since the network is undirected, $i$ is a neighbor at a distance smaller than $r_0$. In (**b**), the size of the giant connected component is a function of $r_0$. The interaction radius is set to $r = 0.1$, above which links cannot be drawn, and kept the same for all simulations. Different system size $N$ are explored, indicated in the legend. Solid lines correspond to the theoretical approximation from Eq. (17) and markers are results from simulations. In (**c**), a discrepancy between theory and simulations, Eq. (18), as a function of the system size. All simulation points are averaged over 100 independent realizations.

the theory due to the topological correlations present in RGGs and not captured in their degree-feature distribution. We can quantify this discrepancy following the rationale of Eq. (13) but taking the absolute value in the integrand. Thus,

$$\epsilon(\mathbf{y}) = \int_{\mathbf{x}_{\min}}^{\mathbf{x}_{\max}} \mathrm{d}\mathbf{x} \left| S_{\text{theor}}(\mathbf{x}, \mathbf{y}) - S_{\text{simu}}(\mathbf{x}, \mathbf{y}) \right|, \qquad (18)$$

where $S_{\text{theor}}(\mathbf{x}, \mathbf{y})$ and $S_{\text{simu}}(\mathbf{x}, \mathbf{y})$ are the analytical expressions of the order parameter given by the theory and obtained via simulations, respectively. Note that a similar expression has been used recently[49]. In order to understand how the size of the network affects the accuracy of the predictions we compute $\epsilon(N)$, see Fig. 4c. We can observe that the discrepancy is reduced as the networks become larger, as already hinted in Fig. 4b. For the range of $N$ explored, the decay trend does not change, hence suggesting that there are not different regimes where our framework works better or fails, but there is a single regime where the effects of topological correlations are gradually washed out as the link density increases.

**Features related to a dynamical process**. As a final application of the feature-enriched percolation, we investigate the case in which the features are coupled to dynamics running on top of the network. There are many situations in which the dynamical evolution of some processes on the network generates some quantity, or attribute, that might not be evenly distributed across nodes. For instance, when studying the problem of synchronization, it is known that in the desynchronized phase, the synchronization error—the time-averaged distance in the phase space between the state of a node and the average state of the system—displays a power-law decrease with the degree[50]. In such a scenario, one might be interested to study what is the surviving network after the removal of the most (or least) synchronized nodes with respect to the mean activity of the system. Another example is found in communication networks when modeling traffic, where nodes with a higher degree tend to be more congested on average[51].

Here we focus on the SIS model, well-known in the context of epidemiology[52]. It consists of nodes that can be in either of two states, susceptible or infected. Infected nodes transmit the disease to susceptible ones at a certain rate upon encounter, and infected nodes recover spontaneously at a different rate. The dynamical evolution of the model is written as

$$\frac{\mathrm{d}x_i}{\mathrm{d}t} = -\tau_1 x_i + \tau_2 \sum_{j=1}^{N} A_{ij}(1 - x_i)x_j, \qquad (19)$$

where $A$ is the adjacency matrix of the network, $\tau_1$ and $\tau_2$ are constants that we set to 1 for convenience and $x_i$ is the probability that node $i$ is infected, hence $x_i \in [0, 1]$. Integrating these equations from a random initial condition, we see that the probability of finding an infected node in the stationary state depends on the node degree. This allows us to define the feature as the probability of infection at the stationary state, i.e., $F_i \equiv x_i(t \to \infty)$. This way, by using the framework of feature-enriched percolation we can study what is the proportion of nodes with the highest probability of ending up being infected need to be removed to dismantle the network. That is, we use again an occupation probability $\phi_F = \theta(-(F - F_0))$, but other choices of course are possible.

For certain dynamical processes on certain types of topologies, one can calculate the exact joint degree-feature distribution and plug it into Eqs. (1)– (5). However, these will be marginal cases over all the ensemble of dynamical models and network architectures, and for many applications, the joint distribution will not be analytically available. To illustrate how one can proceed in this latter case, here we take an agnostic approach and use only information about the node degrees and their feature value to infer an approximate $P(k, F)$. We first collect all the pairs $(k_i, F_i)$ and compute a non-normalized two-dimensional histogram, see Fig. 5a. The collapse in the $k - F$ plane tells us the type of correlation between degree and feature, that for the case of the SIS turns out to be positive. Note that for each value of the degree (recall that $k$ is considered discrete), the distribution $P(F|k)$ has a bell-shape curve, of different height, different mean value, and different width, see the ridgeline plot of Fig. 5b for a better

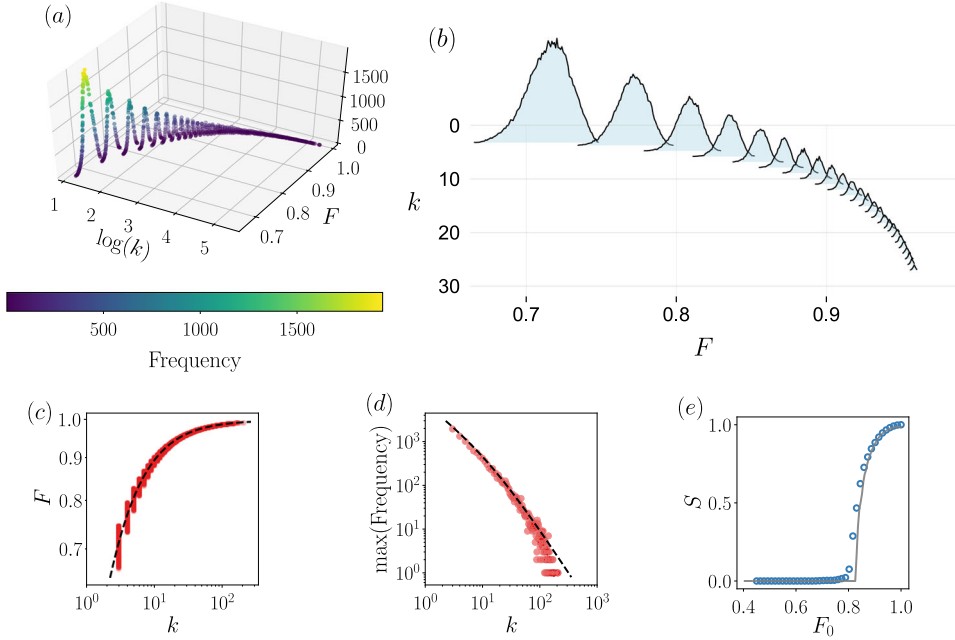

**Fig. 5 Feature-enriched percolation of the SIS model.** We integrate the dynamical Eq. (19) for a reshuffled Barabási–Albert[70], with $m = 3$ and system size $N = 2000$. A non-normalized histogram of the pairs $(k_i, F_i)$, where the feature value is $F_i = x_i(t \to \infty)$, is shown in (**a**). The input data for the histogram comes from 100 independent realizations. In (**b**), ridge plot of the 25 first curves of the histogram, i.e., those with the lowest degree and highest peaks. The bell shape behavior with varying mean, standard deviation, and peak height can clearly be appreciated. In (**c**), projection of the $k - F$ plain, together with the expressions $\mu_F(k)$ (see text) given by the Bayesian machine scientist method. The standard deviation $\sigma_F(k)$ is not shown because it visually overlaps $\mu_F(k)$ for large $k$. The expressions are $\mu_F(k) = \exp(a_1/(b_1 - k))$ and $\sigma_F(k) = \exp(a_2/(b_2 - k)) - \mu_F(k)$, and have been obtained imposing only one free parameter per function, namely $v = -1.1641$ and $w = 0.4317$. The constants in the exponential are given by $a_1 = -v$, $b_1 = -\cosh^v(v^2)$, $a_2 = 2w$ and $b_2 = -\cos^2(w)$. **d** To find the height of the probability peaks we use the BMS on the pairs of variables $(k, \max(P(F|k)))$, indicated by points, obtaining $h(k) = a_3(x(b_3 + x))^{-1}$ where the free parameter is $t = 26.1442$ and the constants in the function are $a_3 = \tan^t(t)/2$ and $b_3 = t/2$, indicated by the solid line. In (**e**), feature-enriched percolation of the SIS model, with the simulation (points) and the theoretical curve obtained from Eq. (21). Each point in (**d**) is averaged over 100 independent realizations of the dynamics.

appreciation. The first strong approximation is to consider that each $P(F|k)$ is proportional to a normal form

$$P(F|k) \propto \exp\left[-\frac{(F - \mu_F(k))^2}{2\sigma_F^2(k)}\right]. \qquad (20)$$

To obtain the values of the mean feature at degree $k$, $\mu_F(k)$, and its standard deviation $\sigma_F(k)$, we employ the Bayesian machine scientist (BMS)[53], a recent algorithm based on Bayesian probability and Monte Carlo Markov chains that it is able to provide the most plausible closed-form expression given a dataset, see Fig. 5c. To incorporate the decaying behavior of the peak height as a function of the degree, we compute the relation between degree $k$ and the maxima of $P(F|k)$ and find the most plausible expression, $h(k)$, again with the BMS, see Fig. 5d. Notice that here we are also assuming that the height of the probabilities do not depend on the feature $F$. In summary, we have an approximate degree-feature distribution

$$P(k, F) = \mathcal{Z} \, h(k) \exp\left[-\frac{(F - \mu_F(k))^2}{2\sigma_F^2(k)}\right], \qquad (21)$$

with $k = k_m, k_m + 1, k_m + 2, \ldots$, and $F \in [0, 1]$, where $k_m$ is a minimum degree seen in the data and $\mathcal{Z}$ is the normalization constant. It is important to note that the joint distribution has been obtained in an unsupervised way and without any prior knowledge of the real degree distribution. In other dynamical processes different from the SIS model the functional expressions used here might not work, but eventually one can always follow similar steps or even apply the BMS to the two-dimensional empirical data to directly compute an approximation to $P(k, F)$.

In Fig. 5e we compare the results of the simulations with the curve obtained from the theory. We have employed the joint degree-feature distribution (21) to compute the generating functions, but we proceed numerically because no closed expressions can be found. We see that the agreement is quite good, despite the strong approximations used during the process. A small discrepancy around the critical region can be appreciated, rooted in finite-size effects and probably a systematic deviation that could potentially be reduced by employing more complicated functions in the process of constructing $P(k, F)$, such as skewed Gaussian distributions. Anyway, we have shown that with little information one can find a quite satisfactory description of the percolation properties of systems in which the features are related to a dynamical process. This works reasonably well for synthetic networks, and we proceed next to test the accuracy of the feature-enriched percolation in real networks, which are characterized by topological correlations not taken into account in the theory.

To this goal, we use three different dynamics on three different real networks and proceed similarly as before, i.e., we will find the approximate degree-feature joint distribution by means of the BMS. The first example is a mutualistic dynamics arisen in symbiotic ecosystems, where the time dependence of the abundance $x_i$ of the species $i$ is given by the equation[54,55]

$$\frac{dx_i}{dt} = \tau_1 x_i(1 - x_i) + \tau_2 \sum_{j=1}^{N} A_{ij} \frac{x_i x_j}{1 + x_j}. \qquad (22)$$

$\tau_1$ and $\tau_2$ are constants, and they will be so in the subsequent models. The first term on the right-hand side models the logistic growth and the second term captures the mutualistic interaction that neighboring species have on species $i$. The network on which

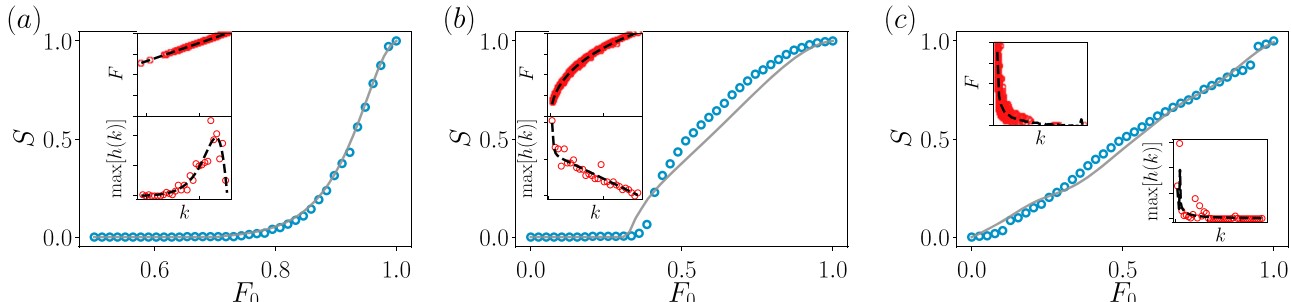

**Fig. 6 Feature-enriched percolation of dynamical processes on top of real-world networks.** In (**a**) mutualistic dynamics are given by Eq. (22), in (**b**) population dynamics from Eq. (23) and in (**c**) the biochemical dynamics (24) The main plots display the size of the largest connected component from simulations (markers) and its approximation given by the theory (lines). Insets show the data used to feed the Bayesian machine scientist (markers) and the resulting curves (dashed lines), used to calculate the degree-feature joint distribution (21). The output functions of the BMS are given in Supplementary Note 8. The feature is the value of the dynamical variable $x_i$ at the steady-state, normalized by its maximum value, i.e., $F = x_i(t \to \infty) / \max_i [x_i(t \to \infty)]$. The range of $F$ in all the insets is the unit interval. The range of $\max[h(k)]$ is arbitrary since its values depend on the binning to compute the histograms. All constants $\{\tau_i\}$ are set equal to 1. The initial condition is a uniform distribution in the unit interval. Each point of the main plot is computed as the average of 100 independent realizations of the dynamics.

we run these dynamics is the one-mode projection[9] of the plant-pollinator bipartite network reported in Ref. [56]. The projection is constructed in such a way that two plants are connected if they are pollinated by the same insect. We apply the feature-based interventions following the step occupation probability that we have been using throughout, i.e., we test the percolation properties of the network when the most abundant plants are removed. The results are shown in Fig. 6a. We observe that the theoretical expressions match very well the size of the largest connected component obtained from the simulations. Notice that the qualitative behavior of the functions used to construct the approximate degree-feature distribution (21) (see insets of Fig. 6a) is different from the previous case and it strongly depends on the dynamics employed. Indeed the feature values grow linearly with the degree while keeping a constant and very small standard deviation, and the peak height is not monotonic with the degree, with its most noisy part located where the peaks are largest.

The second example corresponds to birth-death processes[57]. In the context of population dynamics, the temporal evolution of the population $x_i$ in a site $i$ can be described by

$$\frac{dx_i}{dt} = -\tau_1 x_i^2 + \tau_2 \sum_{j=1}^{N} A_{ij} x_j. \tag{23}$$

The values of the exponents 2 and 1 in the $x_i$ and $x_j$ terms on the right-hand side are arbitrary for the present study, and other choices are of course possible. These particular ones represent pairwise depletion and linear flow between interacting populations, respectively. The dynamics are implemented on top of the one-mode pollinator projection of the previous plant-pollinator network. This network is constructed by connecting the pollinators that pollinate the same plant. The results of the feature-enriched percolation are displayed in Fig. 6b. In this case, we observe that the theory offers estimates for the critical point and for the size of the giant component that are a bit lower than the values given by the simulations. Depending on the application, this discrepancy might be tolerable or not, but anyway, the theoretical calculations are fairly good at catching the response of the system to feature-based interventions.

As a final dynamics, we use the mass-action kinetics model often employed in biochemistry[58]. The equation

$$\frac{dx_i}{dt} = \tau_1 - \tau_2 x_i - \tau_3 \sum_{j=1}^{N} A_{ij} x_i x_j \tag{24}$$

gives the temporal evolution of the concentration $x_i$ of protein $i$. The first term represents the rate at which the protein $i$ is

synthesized, the second term stands for its degradation, and the last term accounts for the interaction between molecules. The real topology where we test the feature-enriched percolation is the *C. elegans* interactome constructed considering the interolog interactions[59]. The results are shown in Fig. 6c, where we find a good agreement between theory and simulations as well.

The dynamical processes presented here are merely illustrative examples of the potential and flexibility of the theory. For all the chosen examples the microscopic rules and the time evolution of the variable of interest were known, but we could have proceeded even without that information. There are two minimal ingredients to apply the feature-enriched percolation: the degree and the feature value for every node. If for whatever reason, a relation between the feature and the degree cannot be obtained, one can always proceed by employing the BMS technique, or other methods whose goal is to provide closed-form equations from data[60–62].

## Discussion

In recent years there has been an upsurge of contributions aiming at offering more realistic descriptions of the natural and socio-technical phenomena that networks encode, e.g., via multilayer[63,64] or temporal[65] networks, or via higher-order interactions[66]. These topological generalizations induce non-trivial consequences in network dynamics, with important implications for the stability and proper functioning of the systems when subjected to perturbations. Beyond these more accurate topological characterizations, there is a dimension that has been frequently ignored in the study of robustness and resilience, which is node metadata. Its omission is not rooted, by no means, in its irrelevance, but because node metadata is a type of information that many times is lost or disregarded in the process of constructing the networked architecture from empirical observations.

Taking percolation as the paradigmatic model to assess the robustness of a network, in this article, we propose a natural generalization of this phenomenon, flexible enough to include node removal protocols based on a combination of the degree and non-topological node metadata, what we call the features. We have worked out the analytical expression for the size of the giant component and have checked its good agreement with simulations. We have discussed in some detail the phenomenology that appeared in a set of examples displaying typical degree-feature relations of real systems, such as the critical exponents of the transition or the characterization of the non-monotonic response

of the robustness induced by the correlations. In this first part of the article, the nature of the features has been left undetermined and, far from being a limitation, this is actually a strength of our model. Indeed, the origin of real node metadata can be either an exogenous or endogenous property of the nodes, can be either of constant or mutable nature, can be either numerical or non-numerical, etc. All these cases can be included in our model. In the second part of the article, we have dealt with two families of problems in which the features have a physical meaning: spatial networks and dynamical processes on networks. The latter case is the most challenging problem because very frequently one cannot analytically extract the main ingredient to use the feature-enriched percolation framework, the joint degree-feature distribution. We have shown, however, how to overcome this limitation by employing a state-of-the-art probabilistic method that gives us an approximate distribution.

A groundbreaking discovery in network robustness assessments was the realization that the response to random failures and degree attacks is radically different when the variance of the degree distribution is much larger than the mean degree. We believe that, in a similar vein, conceptualizing the possibility of attacking networks with new feature-based protocols, and providing the mathematical framework to study this process, can help unravel unexpected responses and hidden fragilities. We hypothesize that it might be possible to choose a smart occupation probability $\phi_{k,F}$ that leads to a truly discontinuous percolation phase transition, thus identifying a crucial subset of nodes (that is, from a network ensemble perspective, identifying the range of degree and feature values) that once removed cause catastrophic consequences for the robustness of the system.

Because our model builds upon traditional, well-grounded message-passing techniques that have been successfully employed in a myriad of different problems, many generalizations not treated in the present work are still possible. Some of them are the study of bond percolation based on features, which is a very relevant situation because many existing network datasets convey information about the link weight rather than node metadata. Generalizations are also possible in topologically correlated networks, i.e., those showing clustering or non-trivial assortativity mixing. Percolation in multilayer networks has also attracted a considerable extent of attention in the last decade, and feature-based protocols can be devised in these layered structures as well. The feature dimension can be relevant too when studying optimal percolation, that is, finding the sets of nodes that, when removed, cause the largest possible reduction in the giant component. Indeed, one can devise attacks that combine feature and topological information, more complex than the one studied in this article, in order to eventually outperform purely topologically-based interventions and move closer to the optimal dismantling. Finally, a conceptually similar problem but that would require a completely different mathematical approach is the one of feature-based percolation in low-dimensional lattices, where it can be addressed which traits of the feature distribution modify the well-known properties of these systems.

Last, but not least, we would like to discuss some implications of our model on the robustness of complex interconnected systems. Firstly, we are opening the doors to the possibility of assessing the behavior of a network as a function of the combination of several traits, enabling the exploration of responses in large phase spaces. This can be particularly useful not only for scientific research problems but for policy making too, where, many times, it is needed to evaluate scenarios taking into account the optimization of a multitude of factors. Think, for instance, in the current COVID-19 pandemic, where policies have required to maintain a delicate balance between the protection of public health and the sustainability of the economic system. Our model represents a first step towards the non-topological multi-dimensional optimization of robustness. Secondly, depending on the nature of the features, different implementations can be devised. There are some situations in which the features values can be tuned at will, e.g., resources are given to certain nodes, therefore it can be explored how to modify the feature allocation, correlating it with topological information, in order to better protect the network. We learned, for instance, that the robustness in networks with positively correlated degree-feature distributions is considerably lower than a network with uncorrelated degree-feature, and that this is not always true for negatively correlated ones. This kind of information can be exploited, of course combining it with attacking protocol. There are other situations in which the features remain fixed, e.g., the age of nodes, but the topology is flexible. Put otherwise, we can tune the strength of correlations at will by redirecting the edges in a convenient way. Here we can also take advantage of the relation between correlation and robustness to increase the system's ability to sustain its function despite the attacks. Finally, we can also use our formalism to infer the temporal evolution of the robustness in the case where the features evolve. For the sake of illustration, in the article, we have presented several examples of dynamics with well-defined equations that display a steady state, but none of these characteristics are actually necessary. It just suffices to have an accurate prediction for the future feature values, whatever the way we have used to obtain it. Taking snapshots at different times, we can apply the BMS method at each of them to obtain a time-dependent size of the largest connected component. This allows us to gain access to information on when a system will be most robust and most vulnerable, hence forestalling undesired behaviors. All these practical examples evince the potential of our framework, from which insightful lessons can be learned to better protect, or dismantle real systems. Likewise, at a fundamental level, a very interesting new phenomenology can be obtained due to the inclusion of features. For all these reasons we hope our model becomes a stepping-stone on the path towards a more realistic and useful description of the process of network robustness.

## Data availability
All datasets used in this article are publicly available on the Internet. The data used in Fig. 1a can be found at https://zenodo.org/record/3553442. The data used in Fig. 1b can be found at https://consonni.dev/datasets/. The data used in Fig. 1c can be found at https://covid19obs.fbk.eu/#/api. The data used in Fig. 6a, b can be found at https://iwdb.nceas.ucsb.edu/html/robertson_1929.html. The data used in Fig. 6c can be found at http://interactome.dfci.harvard.edu/C_elegans/index.php?page=download.

## Code availability
The code for the Bayesian Machine Scientist is available at https://bitbucket.org/rguimera/machine-scientist/. Another code relevant to the paper is available from the authors upon reasonable request.

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

## Author contributions

O.A. performed the analytical computations and the simulations. O.A. and M.D.D. designed the research, discussed and interpreted the results, and wrote and revised the manuscript.

## Competing interests

The authors declare no competing interests.
