## [Peer Review File · Nature Communications]

REVIEWER COMMENTS

Reviewer #1 (Remarks to the Author):

see the attachment (page 3-4 of this file)

Reviewer #2 (Remarks to the Author):

Remarks to the Author:

The manuscript titled 'Percolation on feature-enriched interconnected systems' by Artime and collaborators deals with a novel framework of the percolation transition in networks. Instead of looking at the behavior of "standard" percolation by the degree distribution, the authors focus their attention on the feature-enriched percolation. They developed an analytical expression for the size of the giant component (or the order parameter) by using the generating function method, and then applied their theory into spatial networks and dynamical processes on networks in which the features have physical meanings.

Overall, the work is original and interesting.

There are however several aspects of the paper that require revisions/clarifications. My major concerns are:

(1) The analytical expression for the feature-enriched percolation is indeed based on the standard generating function method which was widely used in percolation theory. For example, Eqs. (1) and (5) are the same as the classical one, the only difference is the occupation probability. The feature enrichment does not add any new information in terms of connectivity. It means that this theory is still heavily relied on the degree distribution. I do not find any novel parts in the theory.

(2) For the percolation process: how to attack the (all) nodes if their features are larger than a given threshold? By randomly or depending on their features values.

(3) For the independent case: I really do not understand why the authors chose the network having an exponential degree distribution [Eq. (6)]. I suggest they use the Erdős-Rényi model or scale free model.

(4) For the universality class: the authors "conclude that the critical properties are the same as the mean-field percolation process, even though if the feature distribution is scale-free." I do not agree this point, since the universality class do not only depend on the degree distribution [see Cohen, et al. Phys. Rev. E 66, 036113] but also the removal or adding progress [see Achlioptas, et al. Science 323, 1453–1455].

Do the authors consider the universality class for the Positively correlated case and Negatively correlated case?

(5) Why the feature is a M dimensional but not N dimensional vector?

(6) In Figure 2 (c), each point is computed by averaging over 100 independent realizations, and the bars are the error bars, am I right? If it is (the error bar is indeed very large, the system size is indeed not large enough), how the authors can obtain the critical exponents with a high precision, for example, 0.347 ± 0.015 .

(7) In Figure 3, it seems that the max value of S is about 0.5. Which means that half of the nodes are not belonged to the largest cluster. Why can it not be reached to 1?

(8) Generally, the generating function method is only valid for an infinite system. I notice that the

authors introduced the system size N into the generating functions [see Eq. (17)]. If N is small, the theory may be wrong. This can be used to explain the inconsistent which is shown in Fig. 4 ($N = 150$).

Minor remarks:

(1) What do the black line stand for in Fig.1 (a-c)?

(2) In page 2, the authors should cite [Physical review E 64 (2), 026118] in the theory (generating function) part.

(3) Before Eq. (22), the authors should cite [Phys. Rev. E 66, 036113].

Reviewer #3 (Remarks to the Author):

Report on "Percolation on feature-enriched interconnected systems" by Oriol Artime and Manlio De Domenico

The paper propose a mathematical framework to dismantle networks combining topological properties and non-topological features.

I find the contribution original and worth publishing, after the author addressed the following issues:

* P1, p3. "This has catastrophic consequences for the security of real-world networks, since most of them display such degree distributions."

> I suggest "several" instead of "most".

* Fig1a (Age vs k): There are two dots at age ~ 0 and at least one dot if at $k \sim 2$ with frequency > 1 .

> If these dots are not mistakes I think they deserve to be commented.

* About the formalism, they define $P(k,F)$ as the density of nodes with degree k and features between $[F,F+dF]$ and $\phi(k,F)$ as the occupation probability of nodes with degree k and features between $[F,F+dF]$.

> I do not see the differences between these two functions.

* The article is new in the sense of taking the feature of nodes into account for the attacks.

In featureless attacks there is vast bibliography on the subject, some of which were already cited in this MS.

> However, I suggest the authors to refer at least to these two articles on standard attacks on general networks

1) Influence maximization in complex networks through optimal percolation
by Morone & Makse, Nature 524, 65 (2015)

The paper reportedly find, analytically, the optimal set to fragment a network according to the collective index (CI).

2) Empirical determination of the optimal attack for fragmentation of modular networks
by de Abreu, Goncalves & da Cunha, Physica A 563, 125486 (2020)

This other addresses the attack of networks in what they call brute force attack, this is finding by exhaustion the set of nodes that reduce the original graph to its smallest component.

It is shown, by the way, that CI is not generally the best strategy.

> I wonder how the proposed formalism could be compared and checked in performance against CI or the brute force approach when it is reduced to networks with the same feature.

In “*Percolation of feature-enriched interconnected systems*”, authors added a new dimension of node for percolation, and called it *feature*. With this additional dimension, authors proposed a math framework to calculate the size of giant component. Based on this framework, authors studied the size of the giant component when feature distribution is independent of degree, positively correlated with degree and negatively correlated with degree. The analytical solutions are given for some cases. Then authors applied their method to random geometric graphs and concluded “for certain values of the link density the feature-based description works reasonably well”. At last authors applied their method to network dynamical process (SIS model on *C. elegans* network). Aided with Bayesian Machine Scientist (to estimate joint probability distribution), authors found that the agreements between theory and simulation are well for eq. 21, eq. 22 and eq. 23.

I appreciate the concept of feature related percolation. My main concern is that although authors proposed a new theoretical percolation framework with nodes features beyond the network structure, they failed to show the advantages of this framework. To be honest, I am not very sure there is any advantage of this framework, because it looks that all the *features related percolation* cases can be solved well using the version of degree-correlated percolation.

So I think this manuscript does not meet the criterion of NC paper. The authors need to show the uniqueness of *feature-enriched percolation*, or demonstrate that *feature-enriched percolation* is superior to degree-correlated percolation in some important way.

Main comments

1) The authors gave a good perspective, but need to put it into practice. They discussed the impact of non-topological features on percolation, and Figs. 1 a-c show that feature distributions are non-trivial. However, the examples of percolation based on real features, as well as the benefits and applications, are not shown. Although authors used *C. elegans* network in section *Features related to a dynamical process*, the results are actual simulated data. They didn't have any empirical study to support their method, therefore the necessity to introduce the new dimension is not very strong. In the three dynamic models of this section, i.e., SIS, birth-death processes and biochemistry, the features are all simulation results, which are determined by topological structure. Therefore, these examples are essential the topology-based model. In short, the idea of the manuscript is very good, but the results are not exciting. I'm expecting that the authors can show some uniqueness of their framework, or demonstrate the *feature-enriched percolation* is superior to the previous percolation theory.

2) The mathematical solution of the feature-enriched percolation is very important for the manuscript, however it is not impressive. The generating function of Eq.2 is the key of the solution, however it can be simplified to $\sum_k P(k) \tilde{\phi}(k) z^k$, where $\tilde{\phi}(k) = \int dF P(F|k) \phi_{k,F}$, meaning the probability that a node with degree k is present in the network. By this simple transformation, the model can be solved in a degree-based percolation. I agree there are some theoretical contributions, however these contributions are not significant.

Other comments

- 1) In page 6 authors point out the existence of “double phase transition as α approaches 1”. It seems to be an interesting and important result. However, the explanation is short and not clear to me. Given that 1 is outside of allowed exponent, a picture of $S(\alpha)$ in the vicinity of 1 is helpful to understand the nature of this double phase transition.

- 2) In case of random geometric graphs (RGGs), authors claim that “tree-like approximations in general are not as accurate as in the case of infinite-dimensional networks” and “for certain values of the link density the feature-based description works reasonably well”. And I think this is an important point to justify their framework. However, authors only gave a figure to show the agreements between simulation and their method. A comparison between their theory and the-state-of-the-art is necessary to back their claim and the limitation is also worth a discussion. For example, what’s the low bound of link density for their method to work?
- 3) Author didn’t discuss the implication of their method for improving robustness of complex system: how can these finds help us to design a better system? In most cases, feature is out of our control (like the age in figure 1) and we can only change the topology of network. When should we do that and why?

Dear Reviewers,

Thank you very much for your insightful comments, which certainly have given us the opportunity to improve the quality of the manuscript. Please find below our response. First we give a list of changes that were not required by you but we deemed necessary to better convey our message and to better organize the structure of the article. Afterwards we offer a point-by-point response to your comments.

List of changes not reported in the point-by-point response

1. We have reformatted the article in order to comply with the Nature Communications' formatting guidelines. The main sections now are Introduction, Results and Discussion, and the old subsection have been redistributed among these 3. We have followed the instructions indicated in <https://www.nature.com/documents/ncomms-formatting-instructions.pdf>.
2. A Supplementary Information file has been included, where we offer a deeper discussion on the universality classes. We also provide some analytical expressions of the models studied and some calculations. We have decided to move the appendices of the old version into this file, as well.
3. Two videos are supplied to support some of the new results reported in the Supplementary Information.
4. Panels of Fig. 2 have been resized to the same dimension and fitted in one row.
5. Figure 4 has been complemented with a sketch that aims to explain the proposed removal mechanism in the context of random geometric graphs (now Fig. 4a).
6. Solved typos in Eq. 12, in Eq. SI17, which was Eq. 27 in the old version, and in the feature definition in the caption of Fig. 6.
7. Several typos corrected, scattered throughout the text.
8. We have added a paragraph at the beginning of the discussion section, in order to make the transition between the Results and Discussion smoother and provide a sort of short introduction to the Discussion.

Response to Reviewer #1

*In “Percolation of feature-enriched interconnected systems”, authors added a new dimension of node for percolation, and called it feature. With this additional dimension, authors proposed a math framework to calculate the size of giant component. Based on this framework, authors studied the size of the giant component when feature distribution is independent of degree, positively correlated with degree and negatively correlated with degree. The analytical solutions are given for some cases. Then authors applied their method to random geometric graphs and concluded “for certain values of the link density the feature-based description works reasonably well”. At last authors applied their method to network dynamical process (SIS model on *C. elegans* network). Aided with Bayesian Machine Scientist (to estimate joint probability distribution), authors found that the agreements between theory and simulation are well for eq. 21, eq. 22 and eq. 23.*

I appreciate the concept of feature related percolation. My main concern is that although authors proposed a new theoretical percolation framework with nodes features beyond the network structure, they failed to show the advantages of this framework. To be honest, I am not very sure there is any advantage of this framework, because it looks that all the features related percolation cases can be solved well using the version of degree-correlated percolation.

We would like to thank the referee for reviewing our paper and for his/her insightful comments. The depth of the comments motivated us to critically revise some parts of our work and better explain why our framework is novel, more general and more useful than existing ones.

So I think this manuscript does not meet the criterion of NC paper. The authors need to show the uniqueness of feature-enriched percolation, or demonstrate that feature-enriched percolation is superior to degree-correlated percolation in some important way.

Main comments

*1)The authors gave a good perspective, but need to put it into practice. They discussed the impact of non-topological features on percolation, and Figs. 1 a-c show that feature distributions are non-trivial. However, the examples of percolation based on real features, as well as the benefits and applications, are not shown. Although authors used *C. elegans* network in section Features related to a dynamical process, the results are*

actual simulated data. They didn't have any empirical study to support their method, therefore the necessity to introduce the new dimension is not very strong. In the three dynamic models of this section, i.e., SIS, birth-death processes and biochemistry, the features are all simulation results, which are determined by topological structure. Therefore, these examples are essential the topology-based model. In short, the idea of the manuscript is very good, but the results are not exciting. I'm expecting that the authors can show some uniqueness of their framework, or demonstrate the feature-enriched percolation is superior to the previous percolation theory.

We thank the reviewer for the comment and for raising these concerns which, of course, deserve to be adequately addressed. To this aim, we would like to answer in two blocks, first referring to the second half of his/her comment, and later to the first part.

The reviewer asks us to show the *uniqueness* or *superiority* of *our framework* with respect to the classical percolation theory. This work is built upon the simple and useful, yet surprisingly overlooked, idea of bringing together for first time topological information and node metadata, i.e., non-topological information, in the study of network robustness. This idea is put forward by (i) justifying its need by looking at non-trivial degree-feature patterns of 3 real networked systems (and proposing there scenarios in which feature-enriched percolation can be relevant), (ii) modeling the phenomenon we aim to describe with easily parsable mathematics, and (iii) giving 8 applications of increasing complexity (3 *ad hoc* $P(k,F)$, RGG, and 4 cases with dynamical features) where the analytical predictions are tested.

It is crucial to remark here that these examples, along with many other applied problems that one can envision, **could not even be conceptualized within previous percolation frameworks**, therefore we do think our article offers uniqueness in that direction. (Added some comments on this in the last paragraph of the Discussion).

Regarding the requested superiority of our framework, it is not clear to us if by “superior” the reviewer is referring to the fact that our framework should be more general than previous percolation frameworks, or if he/she understands “superior” in a quantitative way, where feature-based approaches should be able to faster dismantle the networks, or to better protect the networks. If it is the former, we have already argued that feature-enriched percolation is more flexible than approaches based purely on topological information. If it is the latter, we think that such analyses can be performed but it is difficult to draw general conclusions because, as already shown in some parts, the type and strength of the feature-degree correlation have an enormous impact on the response of the system.

In other words, comparing performances of feature-enriched percolation against classical versions is doable, and surely for certain $P(k,F)$ our framework can outperform

existing methods (at the end, it is a generalization that takes an extra degree of freedom to tune), but we believe that providing such a systematic/general analysis is far from trivial, falls out of the scope of the present work and it would be more suitable for a more technical journal.

The referee also points out that we *do not have any empirical study to support our method*.

First, we would like to highlight that real-world examples are a key piece of the article: the 3 degree-features patterns of Fig. 1 and the 3 real topologies where the dynamics run (not only the *C. elegans*, as the referee points out), Fig. 7. The former is used to motivate our work, **demonstrating that empirical systems display non-trivial patterns worth to be taken into consideration in the percolation process**. The latter is used to show that **our framework actually adapts well in real architectures**, even if $P(k,F)$ is not known or cannot be calculated analytically.

Second, we interpret the referee's comment as he/she would expect an extra application of the theory dealing with **both** empirical topology and empirical features, and we imagine he/she requests it because it can be seen as the most general case where our framework could be applied. If this is so, we have to respectfully disagree with this point of view and in the following we would like to argue why, after briefly introducing our point of view.

In fact, models in physics are useful representations of reality, whose main goal is to accurately reproduce the empirical observations, often at a global scale, by proposing a set of rules among microscopic constituents, often at a local scale. Models have free parameters, as well as initial/boundary conditions, that make them adaptable to describe the same system in a variety of different regimes or scenarios. Assuming right the epistemic principle that all systems can be described, more or less accurately, by models, then we believe it is much more interesting to provide a dynamical model as input for our framework than an empirical system with fixed topology and fixed features which would be rather limited in scope and can be seen as nothing more than particular case of the underlying, general model. Notice that we do not impose any constraints on the dynamical models we have chosen as examples, hence providing a much broader applicability of our framework.

We are confident that the referee will agree with us that a plethora of works, nowadays, published in network science are based on a similar perspective:

- 1) models for scale-free networks are routinely used instead of empirical ones because the underlying power-law connectivity distribution (and its deviations) are well suited to account for some ubiquitous properties of complex systems even if higher-order topological correlation are overlooked by those models;

- 2) stochastic block models are widely used to represent systems exhibiting a mesoscale organization even if those models, as in the previous case, are based on some ubiquitous feature (i.e., modular structure) and neglect many other topological features;
- 3) Small-world models such as Watts-Strogatz's are still used nowadays, after more than 20 years since their introduction, to reproduce another salient feature of complex systems: the existence of shortcuts connecting remote sides of the network thus reducing the average characteristic length of the system, as well as the tendency of real networks to exhibit triadic closure.

Therefore, in comparison with the above cases, we opted to work with the equivalent of a "scale-free model" rather than "a specific empirical scale-free network characterized by a specific scaling exponent and a given amount of global clustering".

We hope that it is more clear that we do not imply that we do not find interesting applying our theory to systems with empirical topology and empirical features. We are certain that this can be done, and we hope that the referee is convinced that our paper provides compelling evidence that our framework can work properly in these cases. However, we believe that such analysis falls out of the scope of a methodological paper like ours – where the main novelty is related to the new framework and its broad applicability – and it is better suited for technical journals, where one can explore in depth the implications of feature-enriched percolation for particular systems in order to draw system-specific conclusions. Since we agree with the reviewer that this is very interesting, we plan to explore this possibility in a future study and we are grateful to the reviewer for raising this point, which now should be better clarified in the revised manuscript.

2)The mathematical solution of the feature-enriched percolation is very important for the manuscript, however it is not impressive. The generating function of Eq.2 is the key of the solution, however it can be simplified to $\sum_k P(k)\bar{\phi}(k)z^k$, where $\bar{\phi}(k) = \int dF P(F|k)\phi_{k,F}$, meaning the probability that a node with degree k is present in the network. By this simple transformation, the model can be solved in a degree-based percolation. I agree there are some theoretical contributions, however these contributions are not significant.

We agree with the referee that his/her change of variable results into the equation of degree-based percolation. However, he/she is considering an indefinite integral, so he/she is missing the fact that the limits of the integral may depend on the feature threshold as well, hence adding an extra dependence on the feature in his/her $\bar{\phi}(k)$ that makes degree-based and feature-based approaches not equivalent.

This lack of equivalence can be seen in the following terms, as well. Mathematically, one cannot recover our equations departing from a description that only takes into account the degree. Put otherwise, **degree-based percolation is a particular case of feature-based percolation**. Beyond mathematical considerations, the two approaches are quite different for the actual physical realization of the process of percolation. Simple targeting protocols based on features, such as the one considered throughout, are mapped to cumbersome removal protocols in degree-based criteria. That is why it is practically impossible to physically obtain/observe the consequences of feature-based percolation directly from degree criteria alone.

To be even more precise, let us use some of the examples studied. The degree-based occupation probability for the simple feature-based occupation probability $\phi_{k,F} = \theta(-(F - F_0))$ in the positively correlated and negatively correlated cases are $\bar{\phi}(k) = (1+k)^{1+\alpha}[(k+1)^{1+\alpha} - (k+F_0)^{1+\alpha}]$ and $\bar{\phi}(k) = [1 - (\frac{kF_0+1}{k+1})^\alpha]^{-1}$, respectively. In our opinion, it is extremely difficult to justify the choice of such occupation probabilities in the context of degree-based percolation if one does not know $\phi_{k,F}$. By extension, it is difficult then to devise a physically meaningful degree-based protocol that reproduces the phenomenology of very simple feature-based protocols.

We hope that the above arguments allow the reviewer to reconsider his concern on this point.

Other comments

1) In page 6 authors point out the existence of “double phase transition as alpha approaches 1”. It seems to be an interesting and important result. However, the explanation is short and not clear to me. Given that 1 is outside of allowed exponent, a picture of $S(a)$ in the vicinity of 1 is helpful to understand the nature of this double phase transition.

We thank the reviewer for motivating us to further explore the critical behavior of the model in the vicinity of $\alpha \sim 1$. We found that there is no scaling whatsoever close to that point, and since the region $\alpha \leq 1$ is not attainable because would lead to a non-normalizable $P(k, F)$, it is not accurate to talk about a double phase transition. We have also offered an alternative derivation of the continuous approach to 0 of the order parameter in this limit, using the generating functions instead of the criticality condition (old Eq. 13), which we believe is much clearer.

We have added in the Supplementary Information the aforementioned analysis, and we have deleted from the main text the references to the double phase transition. Thanks once again for this insightful comment.

2) In case of random geometric graphs (RGGs), author claim that “tree-like approximations in general are not as accurate as in the case of infinite-dimensional networks” and “for certain values of the link density the feature-based description works reasonably well”. And I think this is an important point to justify their framework. However, authors only gave a figure to show the agreements between simulation and their method. A comparison between their theory and the-state-of-the-art is necessary to back their claim and the limitation is also worth a discussion. For example, what’s the low bound of link density for their method to work?

Thank you for the comment. The section of random geometric graphs has been extended by introducing a quantity (Eq. 19 in the new version) that allows us to show that there are no abrupt changes in the error between theory and simulations, but the discrepancy steadily decreases as the link density grows (Fig. 4c). Therefore, a lower bound where our method starts to fail does not exist, in the sense that the convenience for using our framework on RGG will depend on the error (now quantifiable and comparable) each specific application is willing to tolerate.

3) Author didn’t discuss the implication of their method for improving robustness of complex system: how can these finds help us to design a better system? In most cases, feature is out of our control (like the age in figure 1) and we can only change the topology of network. When should we do that and why?

Thank you for pointing out this certainly relevant missing piece of information. We have added the last paragraph in the Discussion section, where we address this issue and we thank you once again for the overall and insightful comments which motivated us to enhance the results and the presentation of our work

Response to Reviewer #2

The manuscript titled 'Percolation on feature-enriched interconnected systems' by Artime and collaborators deals with a novel framework of the percolation transition in networks. Instead of looking at the behavior of "standard" percolation by the degree distribution, the authors focus their attention on the feature-enriched percolation. They developed an analytical expression for the size of the giant component (or the order parameter) by using the generating function method, and then applied their theory into spatial networks and dynamical processes on networks in which the features have physical meanings.

Overall, the work is original and interesting.

We thank the reviewer for his/her positive comments regarding the originality and worthiness of our work.

There are however several aspects of the paper that require revisions/clarifications. My major concerns are:

(1) The analytical expression for the feature-enriched percolation is indeed based on the standard generating function method which was widely used in percolation theory. For example, Eqs. (1) and (5) are the same as the classical one, the only difference is the occupation probability. The feature enrichment does not add any new information in terms of connectivity. It means that this theory is still heavily relied on the degree distribution. I do not find any novel parts in the theory.

We thank the referee for his/her comment. We would like to point out that we agree with the fact that the structure of Eqs. (1) and (5) reminds those found in featureless percolation. This is somehow expected because we rely on the generating function methodology to conduct the calculations. Alternative mathematical frameworks that deal with percolation could have been used, such as [I. Kryven. Bond percolation in coloured and multiplex networks. Nature Communications, 10(1), 1-16 (2019)] or [Hamilton & Pryadko. Tight lower bound for percolation threshold on an infinite graph. PRL, 113(20), 208701 (2014)], and the resulting equations encompassing the feature dimension would resemble to some extent their featureless counterparts as well. That said, we would like to highlight that in this article we are not merely looking for mathematical novelty *per se*, and we do not claim to do so, but rather we believe that our contribution positions itself as a pioneering and systematic exploration of the overlooked relation between topological and non-topological information in the modelling of network robustness.

We respectfully disagree with some parts of the comment, which we would like to clarify. Note that a key and novel quantity is the degree-feature distribution, which is present throughout the article. Taking into account this quantity is a difference with respect to classical percolation. Therefore, we believe that it is not accurate to claim that “*the only difference is the occupation probability*”, since the equations depend on $P(k,F)$, combining in this way the topological and metadata dimensions. For similar reasons, we do not share the statement that “*It means that this theory is still heavily relied on the degree distribution*”. Besides the simplest case of uncorrelated degree and feature, all the other examples of the application of the theory are discussed on the basis of the feature dimension, and its relation with the topology.

(2) For the percolation process: how to attack the (all) nodes if their features are larger than a given threshold? By randomly or depending on their features values.

We are afraid that we are not sure of understanding this comment. However, we are answering what we believe the reviewer is referring to. In case he/she is not satisfied with the answer, we kindly ask him/her to rephrase it, so we can address it properly.

Our framework allows one to consider the most general removal strategy that simultaneously depends on the degree and feature. This is encoded in the occupation probability $\phi_{k,F}$, and represents a generalization of both classical percolation and degree based attacks. The former is recovered by setting $\phi_{k,F} = \phi$, where $\phi \in [0, 1]$. For degree-based attacks, the most frequent scenarios studied in the literature are either the removal of nodes having a degree above or below a certain threshold k_0 , that can be recovered if $\phi_{k,F} = \theta(-(k - k_0))$ or $\phi_{k,F} = \theta(k - k_0)$, respectively. Among the infinite possible choices of occupation probabilities $\phi_{k,F}$, we decided to focus primarily on the one that removes nodes with feature value above a certain threshold, i.e., $\phi_{k,F} = \theta(-(F - F_0))$, and has been chosen for the sake of illustration, exploring its effects in the different settings discussed (random geometric graph, dynamical processes, etc.). Thus, answering the reviewer’s question, with this occupation probability we can attack – i.e., remove – the nodes having a feature larger than F_0 . All the nodes with this condition are removed simultaneously. Note that, given the same undamaged network, the application of this removal protocol will always result in the same perturbed network. Put otherwise, there is nothing stochastic in the removal process (once the topology is given), unlike in classical percolation.

(3) For the independent case: I really do not understand why the authors chose the network having an exponential degree distribution [Eq. (6)]. I suggest they use the Erdős-Rényi model or scale free model.

We thank the reviewer for his/her suggestion. We acknowledge that Erdős-Rényi and scale-free networks are ubiquitous in most network science applications. We decided to use a network with an exponential degree distribution for two main reasons: (i) it is not strange in the study of network percolation (see, e.g. the chapter on Percolation in the famous M. Newman's book *Networks* (2nd edition, 2018)) and, (ii) more importantly, it is an example whose complexity lies between the Erdos-Renyi (too "simple" since all the generating functions are the same) and scale-free models (too "complicated" since the generating functions depend on special functions, and non-trivial behaviors appear depending on the exponent of the degree distribution, which is something we believed that, if included, could obscure the discussion since it is not directly related to features).

For these reasons, we have decided to keep the analysis of the network with exponential degree distribution, and offer in the Supplementary Information file, for the interested reader, an analysis of the two mentioned models, where we give the generating functions and the critical points, and show the good agreement between theory and simulations.

We are confident that the reviewer will appreciate this extension and we thank him/her again for this comment.

(4) For the universality class: the authors "conclude that the critical properties are the same as the mean-field percolation process, even though if the feature distribution is scale-free." I do not agree this point, since the universality class do not only depend on the degree distribution [see Cohen, et al. Phys. Rev. E 66, 036113] but also the removal or adding progress [see Achlioptas, et al. Science 323, 1453–1455].

We completely agree with the referee. The message we wanted to convey here is that power-law degree distributions can change the universality class of percolation, as reported in [Cohen, et al. Phys. Rev. E 66, 036113] (article already cited), but the fact of having power-law feature distributions does not seem to modify the mean-field critical properties (understanding mean-field as the properties observed in lattice percolation for dimensions $d \geq 6$).

This has been clarified at the beginning of the paragraph, when we talk for the first time about mean-field percolation. We have cited [Achlioptas, et al. Science 323, 1453–1455] as well.

Many thanks for this comment, since it allowed us to better explain our point.

Do the authors consider the universality class for the Positively correlated case and Negatively correlated case?

We thank the reviewer for asking this question. We have analyzed the critical behavior of the correlated models and their randomized counterparts, finding that i) there is no critical behavior at $\alpha \sim 1^+$ for the positively correlated case, ii) the positively correlated case and its randomized version belong to the classical percolation universality class, but iii) the negatively correlated case and its randomized version do not. Based on these results, we have removed all the references to the “double phase transition” phenomenon of the positively correlated case.

The analysis has been reported in detail in the Supplementary Information file.

Once again, many thanks for this comment, since it allowed us to better explain our point.

(5) Why the feature is a M dimensional but not N dimensional vector?

M corresponds to the number of features defined in each node, while N is the system size of the network. M and N are totally independent one from the other. The reason to define an M-dimensional vector of features for each node is simply because we aimed to present the theoretical derivations in a setting as general as possible. Note however that from the Applications Section onward, we stick to the case of M=1 for simplicity.

(6) In Figure 2 (c), each point is computed by averaging over 100 independent realizations, and the bars are the error bars, am I right? If it is (the error bar is indeed very large, the system size is indeed not large enough), how the authors can obtain the critical exponents with a high precision, for example, 0.347+-0.015.

Exactly, for each system size we compute 100 independent realizations, resulting in the marker (average value) and the error bar. However, the deviation from the mean value (i.e. the size of the error bar) is barely dependent on the number of realizations, as far as the sample is high enough. The deviation is related, though, with the stochastic nature of the process, that induces normally distributed values of S around the mean value. The error bars are not symmetric because of the log-log axes.

We acknowledge that proper critical exponent fitting is a quite hard task. One must be very careful on how to do it, especially when the goal is to obtain critical exponents with high accuracy, as some strands of research do when analytical predictions are not available (e.g., in the three-dimensional Ising model). In our case, the goal is more modest: to verify that feature-based percolation in networks with uncorrelated degree and feature belongs to the ordinary percolation universality class.

Following that, at the critical point, $S(a, N) = N^{-\beta_a/\nu_a}$ and $S(\alpha, N) = N^{-\beta_\alpha/\nu_\alpha}$, we take the logarithm on both sides of the relations and fit a straight line to the resulting values. The slope is the ratio of critical exponents, and its error, indicated in the paper, it is the error

of the slope. They have been obtained with the ordinary least square (OLS) method and its covariance matrix, respectively.

Let us explain the rationale behind our procedure, which is based on two reasons. The first one is statistical: OLS assumes homoscedasticity (same variance for all points in the graph), which is approximately true in our data. When one is far from this regime (heteroscedasticity), there are several ways to fit the data and to obtain the errors: bootstrap methods [M. R. Chernick, *Bootstrap methods: A guide for practitioners and researchers*. John Wiley & Sons (2011)], weighted least square algorithms [M. Krystek, & M. Anton, *Measurement Science and Technology*, 18(11), 3438 (2007)], etc. The heteroscedasticity of our data could lead to larger estimates of the error, and this is related with the second reason we were mentioning, which is more physical. Since our objective is to verify the universality class, it suffices to fit the data with a method that can produce estimates smaller than the one that takes into account more assumptions (heteroscedasticity, asymmetric errors, etc.). If, with our more restrictive method, we verify the hypothesis we are looking for, then there is no need to fit with more complicated, subtle methods. We believe that the level of description given in the article is enough to convince the reader that we are dealing with the percolation universality class.

An alternative way to proceed, which we report in the Supplementary Information, is to compute the order parameter for different system sizes and apply to them finite-size scaling relations in order to collapse the curves. We show that using the mean-field percolation exponents the curves collapse well, verifying the results that we reported in the first version of the article.

We would like to thank the referee once again for this valuable comment.

(7) In Figure 3, it seems that the max value of S is about 0.5. Which means that half of the nodes are not belonged to the largest cluster. Why can it not be reached to 1?

In Fig. 3 it is shown the dependence of the giant component S as a function of α , the exponent of the degree-feature distributions introduced in Eqs. 10 and 14. The removal of nodes, however, it is based on their feature value according to $\phi_{k,F} = \theta(-(F - F_0))$ with $F_0 = 3$, as indicated in the caption. Nothing guarantees that, under this removal, S should reach 1 for a certain value of α , since we are always removing those nodes with a feature value greater than 3. Actually, S would be 1 only in two (related) cases: *i)* F_0 is larger than the maximum allowed feature, or *ii)* $p(F)$ is cut off at F_0 .

The observed maximum value of S, close to 0.5, is circumstantial to the value $F_0 = 3$, and changes when other values of F_0 are used. To highlight that not only the maximum but the entire curve changes with the feature threshold, and hence the effect induced by the correlations might grow or diminish, we have added insets in both panels of Fig. 3 to better understand the role of F_0 . This is related to a newly introduced quantity, Eq. 13 in the new version.

(8) Generally, the generating function method is only valid for an infinite system. I notice that the authors introduced the system size N into the generating functions [see Eq. (17)]. If N is small, the theory may be wrong. This can be used to explain the inconsistent which is shown in Fig. 4 ($N = 150$).

We could not agree more with the comment of the reviewer. The introduction of the “system size” (between quotation marks because it should not be understood as the system size in networks not embedded in physical spaces, but more like a link density) in the generating functions is made by setting the maximum degree that a node can have, $N - 1$, and cutting the sums (Eqs. 2 and 4) at that value. This is necessary because we are dealing with spatial networks in bounded domains, hence the thermodynamic limit $N \rightarrow \infty$ of the graph is ill-defined due to this boundedness, as far as the interaction radius r is finite.

We have added a new panel in Fig. 4, in order to address the accuracy of the results as a function of the system size. As pointed out by the referee, the general tendency is to observe larger discrepancies between theory and simulations as N decreases. We discuss this behavior in the text. This analysis is related to the newly introduced quantity in Eq. 19.

Minor remarks:

(1) What do the black line stand for in Fig.1 (a-c)?

We want to thank the referee for identifying this missing piece of information. The black line corresponds to the mean value of the data as a function of the degree. This information has been added in the caption of Figure 1.

(2) In page 2, the authors should cite [Physical review E 64 (2), 026118] in the theory (generating function) part.

We thank the reviewer for pointing out this seminal article of Newman and collaborators that was oversought in the first version. It is cited in the new version in the *The Model* section.

(3) Before Eq. (22), the authors should cite [Phys. Rev. E 66, 036113].

Thank you for the suggestion. We believe, though, that in this raised point he/she made a mistake with the Equation number. Eq. 22 gave the time evolution of the state variable $x(t)$ for the population dynamics model, while the suggested article [Phys. Rev. E 66,

036113] deals with the calculation of the critical exponents of scale-free networks as a function of the degree exponent.

The paper that the reviewer suggests was already cited in the Introduction, but we imagine that here the reviewer is maybe referring to Eq. 24 in the appendix (old version). We have added a citation to the suggested paper right before Eq. 24 (now in the new version it is Eq. SI-14), but we kindly ask the reviewer to let us know if he/she was considering another part of the article for the citation of [Phys. Rev. E 66, 036113].

Response to Reviewer #3

The paper propose a mathematical framework to dismantle networks combining topological properties and non-topological features. I find the contribution original and worth publishing, after the author addressed the following issues:

We thank the reviewer for the time devoted in reviewing our work and for his/her positive comments regarding its originality and worthiness.

** P1, p3. "This has catastrophic consequences for the security of real-world networks, since most of them display such degree distributions."*

> I suggest "several" instead of "most".

We thank the referee for this suggestion.

We acknowledge that there is an active and on-going debate on whether real networks are scale-free or not. The most constricted criterion, based on statistical tests [Broido & Clauset. (2019). Scale-free networks are rare. Nature Communications, 10(1), 1-10], finds that up to 75% of the large number of networks analyzed in the paper show some level of scale-freeness. Networks with long-tailed distributions, the ones we are talking about in the paragraph, can be considered as displaying some level of scale-freeness. On this basis we believe that using "several" could be misleading, since it can be interpreted as "more than two but fewer than many". Instead of "most", though, that we agree that could be interpreted as an overestimation, we have used "many". We have cited the article of Broido & Clauset to offer more context.

** Fig1a (Age vs k): There are two dots at age ~ 0 and at least one dot if at $k \sim 2$ with frequency > 1 .*

> If these dots are not mistakes I think they deserve to be commented.

We confirm that these points are not a mistake from our side and that they do appear in the dataset that we downloaded. We have contacted the curator of the dataset (Anna Evtushenko) asking if she had any clue about the origin of these points. Her response is that the points were present as well in the raw data she used to construct the dataset, but she believes they are a bug and removed them from her analyses.

We have proceeded similarly and have removed them. There were so few of them, in comparison to the total number of directors (164 out of 195615), that their removal is barely perceptible in the mean curve.

** About the formalism, they define $P(k,F)$ as the density of nodes with degree k and features between $[F,F+dF]$ and $\phi(k,F)$ as the occupation probability of nodes with degree k and features between $[F,F+dF]$.*

> I do not see the differences between these two functions.

The joint degree-feature distribution $P(k,F)$ is the probability that a node has k connections to other nodes in the networks, and an internal feature with value within the interval $[F, F + dF]$. On the other hand, the occupation probability $\phi(k,F)$ is the probability that a node with degree k and feature value F has not been removed from the network. Their main difference is that the former is a probability density and needs to be normalized to unity, while the latter is just a two-variable function with the only requirement that its image lies within the unit interval. The occupation probability $\phi(k,F)$ can be seen as the proportionality factor between the original degree-feature distribution of the original network and the degree-feature distribution of the percolated structure.

We have clarified in the text that $\phi(k,F)$ corresponds to the probability that a node has not been removed from the original network, many thanks for pointing this out.

** The article is new in the sense of taking the feature of nodes into account for the attacks. In featureless attacks there is vast bibliography on the subject, some of which were already cited in this MS.*

> However, I suggest the authors to refer at least to these two articles on standard attacks on general networks

1) Influence maximization in complex networks through optimal percolation by Morone & Makse, Nature 524, 65 (2015) The paper reportedly find, analytically, the optimal set to fragment a network according to the collective index (CI).

2) Empirical determination of the optimal attack for fragmentation of modular networks by de Abreu, Goncalves & da Cunha, Physica A 563, 125486 (2020) This other addresses the attack of networks in what they call brute force attack, this is finding by exhaustion the set of nodes that reduce the original graph to its smallest component. It is shown, by the way, that CI is not generally the best strategy.

Thank you for the suggestions. We have included both references in the text.

> I wonder how the proposed formalism could be compared and checked in performance against CI or the brute force approach. when it is reduced to networks with the same feature.

Certainly, taking into account the node features opens the door to devise new attack strategies that can be systematically compared against the metric of Collective Influence, the brute force approach, or any other removal protocol based on topological descriptors. Regarding the comparison of performances, we believe that it is hard to draw general statements about how feature-based attacks compare to purely topological-based ones since the type of degree-feature correlation will significantly influence the response of the system. However, we would like to notice that in a similar way to what it is done for featureless networks, the quest for the optimal set of nodes is valid in feature-enriched systems as well. Actually, combining the information of which nodes are in the optimal set and what are the values in the feature space of these nodes can lead to a better understanding of the robustness and resilience of empirical systems beyond their topology.

We have added a short reference to optimal percolation and to the performance of the feature-based attacks in the discussion section.

REVIEWERS' COMMENTS:

Reviewer #1 (Remarks to the Author):

According to the authors' response, I reevaluate this manuscript and think it could be published in NC. The authors show some advantages of the feature-enriched percolation in dealing with percolation problems, and the feature-enriched percolation also has intuitive physical picture. It looks that it is much suitable to deal with data-driven percolation model.

Hope the following suggestion could be useful for the author.

Maybe it is good to put the additional discussion on independent case in the SI, because independent case is essentially equivalent to randomly removing node, and the relevant parameters only control the number of nodes removed. It is a little bit trivial. There is no need to discussion in the main text.

Fig5 a and b are redundant and perhaps only Fig. 5a is enough.

Regarding Fig1Bottom, the three network figures are very similar. Especially the last two figures, I cannot see any distinct difference between each other if we swap positions.

Fig2c, '(c)' should be outside the figure box.

Reviewer #3 (Remarks to the Author):

The author have solved all the issues I pointed out in the first version, so I recommend the publication for the present MS.

Reviewer #1:

According to the authors' response, I reevaluate this manuscript and think it could be published in NC. The authors show some advantages of the feature-enriched percolation in dealing with percolation problems, and the feature-enriched percolation also has intuitive physical picture. It looks that it is much suitable to deal with data-driven percolation model.

We would like to thank the reviewer for his/her time in reviewing our manuscript and for his/her positive comments.

Hope the following suggestion could be useful for the author.

Maybe it is good to put the additional discussion on independent case in the SI, because independent case is essentially equivalent to randomly removing node, and the relevant parameters only control the number of nodes removed. It is a little bit trivial. There is no need to discussion in the main text.

Fig5 a and b are redundant and perhaps only Fig. 5a is enough.

Regarding Fig1Bottom, the three network figures are very similar. Especially the last two figures, I cannot see any distinct difference between each other if we swap positions.

Fig2c, '(c)' should be outside the figure box.

Thank you for the suggestions. We have decided to implement some of them. For the ones that we have decided to keep, in the following we justify our reasons. Going point-by-point:

- We have modified the bottom panel of Fig. 1, so now the 'Degree' and the 'Feature' cases display completely different deleted nodes, while before two out of the four deleted nodes were shared.
- We have placed the '(c)' outside the frame in Fig. 2c.
- Regarding the Fig. 5b, we agree that it shows the same data as Fig. 5a, but we believe that offers a much clearer detail of the shapes of the curves for degrees that are not small ($k \geq 10$), needed to justify the Gaussian approximation of Eq. 21. We think that it is a good idea to keep it.
- Regarding the suggestion of transferring the "Independent case" section to the SI file, we agree that, at practical effects, independent degree-feature distributions are very similar to random percolation. Despite this fact, we believe it is helpful to provide this first simple example instead of directly starting with the positively and negatively correlated cases because they are more difficult to interpret: generating functions depending on special functions, no closed expressions for the critical points, non-trivial critical behavior, etc. Therefore, we think that for the general reader of Nature Communications the presence of that part in the main text is beneficial.

Reviewer #3 (Remarks to the Author):

The author have solved all the issues I pointed out in the first version, so I recommend the publication for the present MS.

Thank you to the reviewer for his/her time devoted to revising our manuscript and for the positive recommendation.